# Optimized fragmentation schemes and data analysis strategies for proteome-wide cross-link identification

Fan Liu[1,2,*,†], Philip Lössl[1,2,*], Richard Scheltema[1,2], Rosa Viner[3] & Albert J.R. Heck[1,2]

We describe optimized fragmentation schemes and data analysis strategies substantially enhancing the depth and accuracy in identifying protein cross-links using non-restricted whole proteome databases. These include a novel hybrid data acquisition strategy to sequence cross-links at both MS2 and MS3 level and a new algorithmic design XlinkX v2.0 for data analysis. As proof-of-concept we investigated proteome-wide protein interactions in *E. coli* and HeLa cell lysates, respectively, identifying 1,158 and 3,301 unique cross-links at ~1% false discovery rate. These protein interaction repositories provide meaningful structural information on many endogenous macromolecular assemblies, as we showcase on several protein complexes involved in translation, protein folding and carbohydrate metabolism.

[1] Biomolecular Mass Spectrometry and Proteomics, Bijvoet Centre for Biomolecular Research and Utrecht Institute for Pharmaceutical Sciences, University of Utrecht, Padualaan 8, 3584 CH Utrecht, The Netherlands. [2] Netherlands Proteomics Center, Padualaan 8, 3584 CH Utrecht, The Netherlands. [3] Thermo Fisher Scientific, 355 River Oaks Parkway, San Jose, California 95314, USA. † Present address: Leibniz-Institut für Molekulare Pharmakologie (FMP), Robert-Rössle-Str. 10, D-13125 Berlin, Germany. * These authors contributed equally to this work. Correspondence and requests for materials should be addressed to F.L. (email: F.liu@uu.nl) or to A.J.R.H. (email: a.j.r.heck@uu.nl).

Chemical cross-linking combined with mass spectrometry (XL-MS) has emerged as a powerful approach to investigate protein conformations, as well as protein–protein interactions. In XL-MS, proteins are first covalently conjugated with XL reagents and subsequently proteolytically digested into cross-linked peptides before MS identification. These experiments confer the spatial proximity (that is, their maximum distance as defined by the cross-linker spacer arm length) of the two linked residues. From the identified distance restraints, structural information of proteins and protein complexes can be deduced[1–5].

Although seemingly straightforward, cross-link identification in XL-MS studies has been considered very challenging. In conventional bottom-up proteomics approaches, linear peptides are identified based on the accurate precursor mass of the intact peptide (measured at the MS1 level) and sequence-specific fragment ions (measured at the MS2 level). However, this precursor and fragment ion association is impaired in cross-link identification, because the cross-linked peptide is composed of two covalently connected linear peptides. This special structural feature causes two problems during peptide fragmentation and database searches. First, the search space is quadratically expanded, because the search engine needs to consider all possible pairs of linear peptides that match the mass of the cross-linked precursor. Second, the two covalently conjugated peptides are co-fragmented in the same MS2 spectrum, often compromising the fragmentation efficiency of either of the two linked peptides thus hampering identification[6,7]. To address these challenges, MS-cleavable cross-linkers have been developed[8]. These reagents are cleaved in the mass spectrometer, enabling gas phase dissociation of the cross-linked peptides hence greatly facilitating cross-link identification.

In our recent study, we developed an integrated workflow that combines MS-cleavable cross-linkers, a dual fragmentation strategy employing sequential collision-induced dissociation (CID) MS2 and electron transfer dissociation (ETD) MS2 and the dedicated search engine XlinkX, to unambiguously identify cross-links against full proteome databases[9]. In this approach, we successfully overcame the above mentioned database expansion problem by retrieving the precursor mass of each linked peptide based on the fragmentation pattern of MS-cleavable cross-linkers that gives rise to signature peaks with a unique mass difference ($\Delta m$) in the CID–MS2 spectrum. We also significantly improved the quality of fragment ion spectra by performing ETD–MS2 on the same cross-linked precursor, rendering additional sequence-specific fragment ions. This workflow allowed us to identify more than two thousand cross-links from a whole HeLa cell lysate, illustrating its applicability for proteome-wide XL studies.

At the moment, the XlinkX strategy critically depends on high quality CID–MS2 spectra containing all signature peaks of the MS-cleavable cross-linker. Moreover, it requires ETD–MS2 spectra to confidently sequence the two linked peptide constituents. These two prerequisites limit its application to ETD-enabled instruments and a small set of MS-cleavable cross-linkers that provide high quality signature peaks upon CID or higher-energy collision dissociation fragmentation. To overcome these limitations, we developed XlinkX v2.0, implementing several novel MS acquisition strategies and data analysis algorithms (Fig. 1). XlinkX v2.0 supports a hybrid MS2–MS3 fragmentation approach for Orbitrap Fusion/Lumos instruments, which mitigates the necessity of ETD fragmentation. Moreover, it provides an intensity-based precursor mass determination strategy, enabling the identification of cross-links with non-ideal fragmentation patterns. Of note, the described strategies are generally applicable to any MS-cleavable cross-linker presenting unique fragmentation patterns, such as DSSO[10], BrUrBr[11] and SuDP[12]. Collectively, these novel features increase the versatility of the XlinkX v2.0 workflow and improve the cross-link identification confidence, as we demonstrate by applying this improved XL-MS workflow to cross-linked *Escherichia coli* and human (HeLa) lysates. Both proteome-wide XL-MS studies reveal thousands of high-confidence cross-links, allowing us to structurally characterize the conformations and interactions of various endogenous protein complexes.

## Results

**Hybrid MS2–MS3 fragmentation strategy**. MS-cleavable cross-linkers, such as DSSO, have been introduced to facilitate cross-link identification by generating signature fragmentation patterns in the CID–MS2 spectrum[10–13]. In the initial study of the DSSO cross-linker, these signature peaks were selected for MS3 acquisitions to obtain peptide sequence information[10]. Alternatively, cross-links can also be directly identified from MS2 spectra, as described in our previously published algorithm XlinkX[9], enabling MS acquisitions with faster duty cycles and higher sensitivity. As MS2 fragments the entire cross-link while MS3 fragments each linked peptide moiety individually, both acquisition levels provide unique sequence specific fragment ions. Exploiting the benefits of these diverse fragmentation behaviors, XlinkX v2.0 is capable of combining the sequence information from both MS2 and MS3 levels to achieve a more comprehensive sequence coverage of the cross-linked peptides.

As a proof-of-concept, we fractionated a cross-linked *E. coli* cell lysate by strong cation exchange chromatography (SCX) and performed preliminary liquid chromatography–tandem MS (LC–MS/MS) experiments on a subset of *E. coli* SCX fractions (6 out of 20 fractions), acquired with 4 different MS acquisition strategies, that is, (1) CID–MS2, (2) CID–ETD–MS2 (sequential CID–MS2 and ETD–MS2 acquisitions on the same MS precursor ions), (3) CID–MS2–MS3 (CID–MS3 scans targeting each of the signature peaks formed in CID–MS2) and (4) CID–MS2–MS3–ETD–MS2 (a combination of 2 and 3). For the targeted MS3 scans, we made use of the Orbitrap Tribrid instruments (Orbitrap Fusion and Lumos), which allow MS3 acquisitions to be triggered by a specific mass difference. In this way, MS3 can specifically target the two pairs of cross-linker-cleaved signature peaks due to their unique mass difference (Fig. 1). In comparison with conventional intensity-triggered MS3 acquisitions, this mass difference-triggered MS3 significantly increases the success rate of targeting both pairs of cross-linker modified intact peptides for sequencing.

Analysing six *E. coli* SCX fractions with the above described MS acquisition strategies yielded 144 (CID–MS2), 373 (CID–ETD–MS2), 424 (CID–MS2–MS3) and 498 (CID–MS2–MS3–ETD–MS2) cross-links using a 2% false discovery rate (FDR) (Fig. 2a and Supplementary Fig. 1a). Strikingly, we obtained nearly three times more cross-links by supplementing MS3 (CID–MS2–MS3) compared with the CID–MS2 strategy, highlighting the benefit of the MS3 approach. Furthermore, CID–MS2–MS3 acquisitions provide a 14% increase on cross-link identification compared to the CID–ETD–MS2 strategy presented in XlinkX v1.0 (ref. 9). This result shows that the availability of MS3 in XlinkX v2.0 alleviates the dependency on ETD, allowing in-depth cross-link identification even when ETD is not used/available. Lastly, the most hybridized strategy CID–MS2–MS3–ETD–MS2 provides the highest number of identifications among all four acquisition strategies, revealing 34% more unique cross-links than the initial XlinkX approach CID–ETD–MS2. These results illustrate the benefit of supplementing additional fragmentation strategies to enhance cross-link identification.

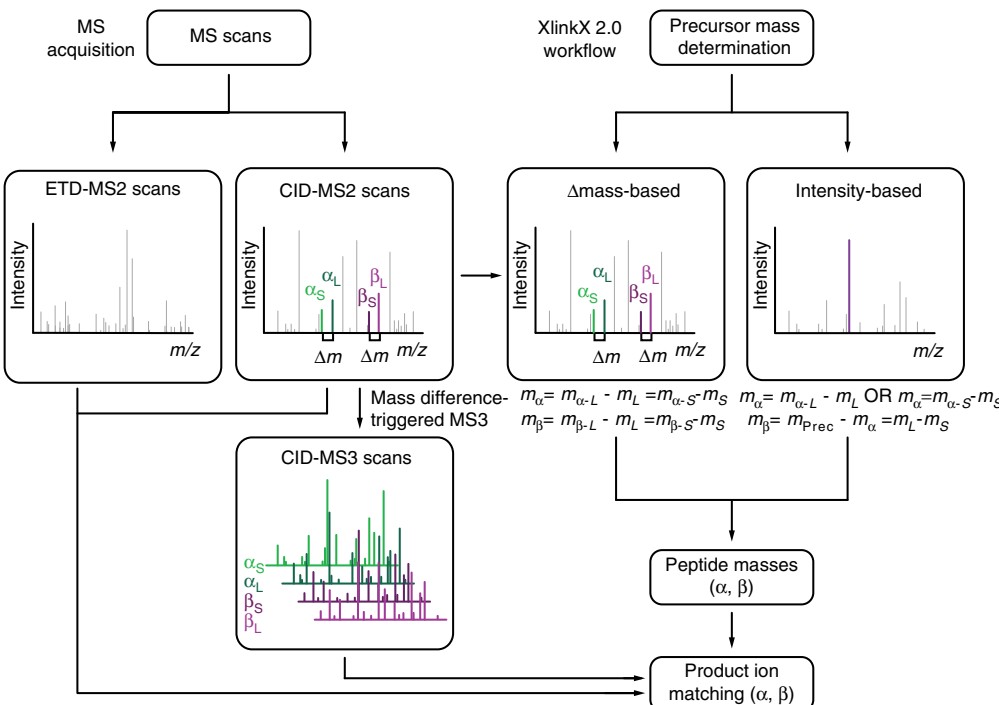

**Figure 1 | The multi-dimensional XL-MS data acquisition and analysis strategy.** Each MS1 precursor ion is subjected to sequential CID–MS2 and ETD–MS2 fragmentation. Data-dependent MS3 scans are performed if a unique mass difference ($\Delta m$) is found in the CID–MS2 scans. In XlinkX v2.0 data analysis, CID–MS2 scans are used to calculate the potential precursor mass of each linked peptide, using $\Delta m$-based, intensity-based or both strategies (see Methods). Both CID–MS2 and ETD–MS2 scans, as well as MS3 scans, are subjected to product ion matching to sequence the two peptide constituents of a cross-link. The four signature fragment ions with a unique mass difference ($\Delta m$), resulting from CID-induced cross-linker cleavage are shown in colours.

In addition, we resembled the MS3-only approach of the initial DSSO XL-MS study[10], searching our CID–MS2–MS3 data based on only the MS3 spectra originating from each of the cross-linker modified intact peptides while ignoring the MS2 fragment ion information (see Methods). As a result, the number of cross-link identifications dropped from 424 to 130, demonstrating that the high-mass-accuracy peptide fragment ions present in the CID–MS2 spectra greatly enhance the efficiency and confidence of the cross-link identification (Fig. 2a and Supplementary Fig. 1b).

**Intensity-based precursor mass determination strategy.** In the previous version of XlinkX, the precursor mass of each linked peptide was determined based on the masses of all four cross-linker-cleaved ions, which requires the presence of all signature peaks in the CID-MS2 spectrum (referred to as $\Delta m$-based approach). If any of the four signature peaks was missing, XlinkX would omit the spectrum from downstream search, thus neglecting cross-links with a less optimal fragmentation behavior. To overcome this limitation, XlinkX v2.0 provides, next to the $\Delta m$-based approach, an intensity-based mass determination strategy. In this approach, we calculate the precursor mass of each linked peptide based on the masses of the cross-linked precursor and one signature peak (Fig. 1 and Methods). Consequently, only one of the four signature peaks needs to be among the top $n$ most intense ions in the CID–MS2 spectrum, to calculate the precursor masses, whereby $n$ is a user defined parameter. We chose $n = 3$ in this study, which was empirically determined based on the intensity ranking of the signature peaks (Supplementary Discussion and Supplementary Fig. 2).

To investigate the applicability of the intensity-based precursor mass determination strategy, we searched the aforementioned six cross-linked E. coli SCX fractions (acquired in CID–MS2–MS3–ETD–MS2 strategy) with both $\Delta m$-based and intensity-based approach, respectively, identifying 498 and 525 cross-links (Fig. 2a). Example spectra for the two approaches are shown in Fig. 2b,c. The overlap between the two methods is very high, as the DSSO crosslinker fragments very efficiently under CID conditions, mostly generating four highly intense cross-linker-cleaved signature peaks. However, both precursor determination strategies provide unique cross-link identifications, cumulatively yielding 575 cross-links from 6 E. coli fractions (Fig. 2a). The intensity-based approach of XlinkX v2.0 uniquely revealed 77 cross-links that would have been omitted in the previous XlinkX version, because any of the 4 signature peaks was missing. Therefore, we expect the intensity-based strategy to be particularly useful for XL-MS studies employing cleavable cross-linkers that have a higher variation of the signature fragmentation pattern, such as cross-linkers with multi-functional groups (for example, affinity-tagged cross-linkers) or multiple cleavable sites. Notably, 50 cross-links were only detected with the $\Delta m$-based approach but not with the intensity-based approach. These cross-links are attributable to MS2 spectra that contain all four signature peaks, while none of them is among the top $n$ most intense ions. We conclude that combining the two search strategies will generate the most comprehensive XL data sets.

**Proteome-wide XL-MS studies of E. coli and HeLa cell lysates.** To demonstrate the full potential of the XlinkX v2.0 workflow, we performed comprehensive XL-MS studies on E. coli and human (HeLa) cell lysates. The data were acquired using our best performing CID–MS2–MS3–ETD–MS2 strategy and the data analysis was performed by XlinkX v2.0 using the combined $\Delta m$-based and intensity-based search mode. Applying XlinkX v2.0 on a whole E. coli lysate yielded 1,158 unique Lys–Lys cross-

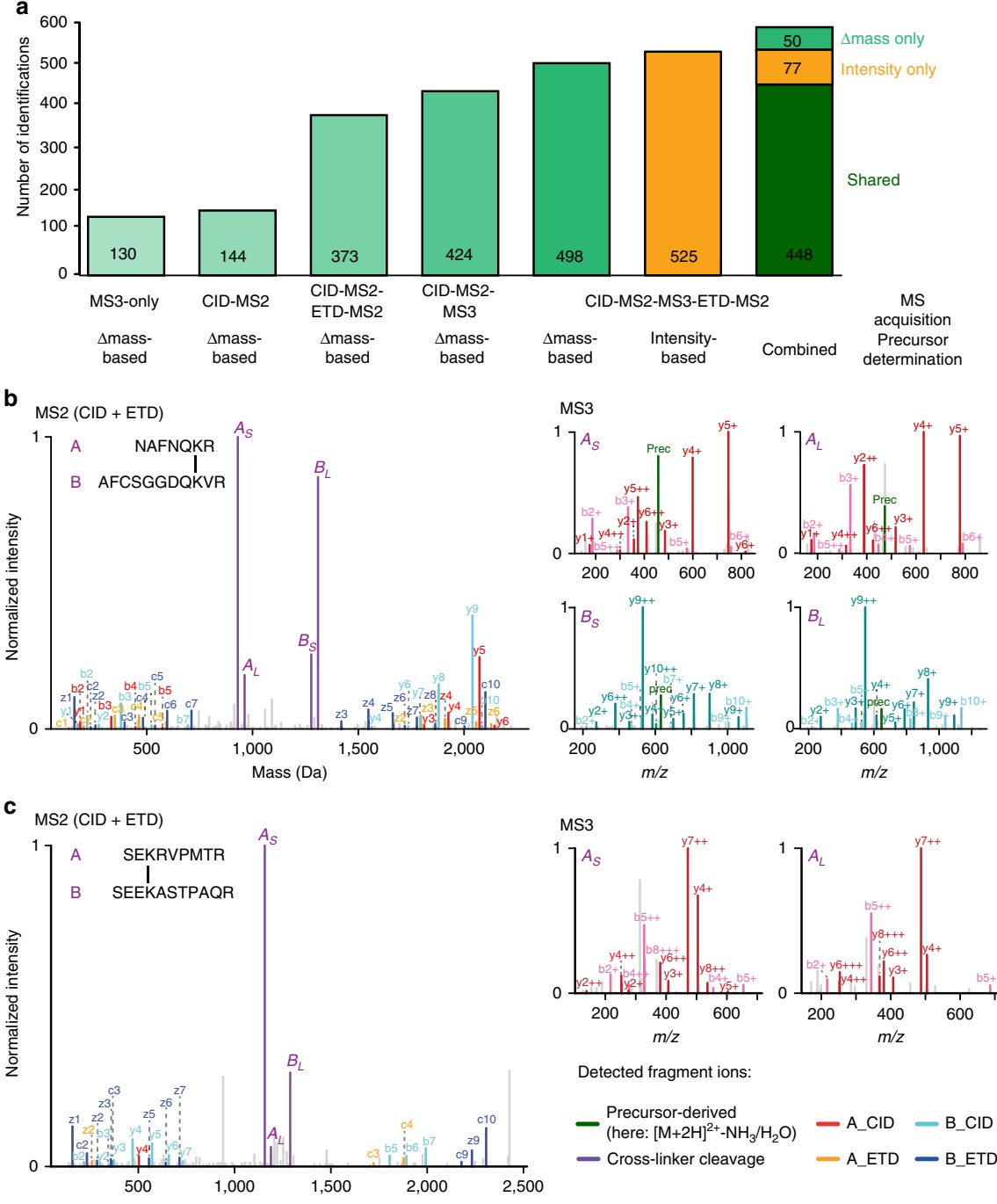

**Figure 2 | Benchmarking different MS acquisition and XlinkX v2.0 data analysis strategies using six *E. coli* SCX fractions.** (**a**) The total number of identified crosslinks using different MS acquisition strategies and/or different precursor mass determination strategies. (**b**) Example spectra of a crosslink identification from the hybrid CID–MS2–MS3–ETD–MS2 strategy. The identified cross-link is formed between Lys-89 and Lys-273 of *E. coli* 1,4-dihydroxy-2-naphthoyl-CoA synthase. (**c**) Example spectra of a cross-link identification from the intensity-based search mode. Only three ($A_s$, $A_L$ and $B_s$) out of four signature peaks are detected and, therefore, MS3 is only triggered for one of the linked peptides. The cross-link between α-ketoglutarate dehydrogenase ODO2 Lys-94 and Lys-177 can still be unambiguously identified, as ETD–MS2 and CID–MS2 data allow for confident sequencing of the *B* peptide.

links at ∼1% FDR and 1,330 cross-links at ∼2% FDR (Supplementary Data 1 and 2). To test the validity of our data, we mapped a large part of our cross-links onto high-resolution structures of translation machineries (Fig. 3a–c and Supplementary Fig. 3a,b), chaperone protein complexes (Fig. 3a), DNA-binding proteins (Fig. 3d and Supplementary Fig. 3c) and carbohydrate degrading enzymes (Fig. 4). This structural analysis is detailed in the Supplementary Discussion. As one of the

highlights, our XL-MS data covered three major *E. coli* chaperone systems; the trigger factor (TF), DnaK and the GroES–GroEL complex. The identified cross-links revealed direct contacts between ribosomal subunits and the DnaK and TF chaperone systems. Specifically, we identified two cross-links between TF and ribosome (TF–ribosomal protein L23 and TF–ribosomal protein L24), and one cross-link between DnaK and ribosomal protein L25 (Fig. 3a), providing further evidence for co-

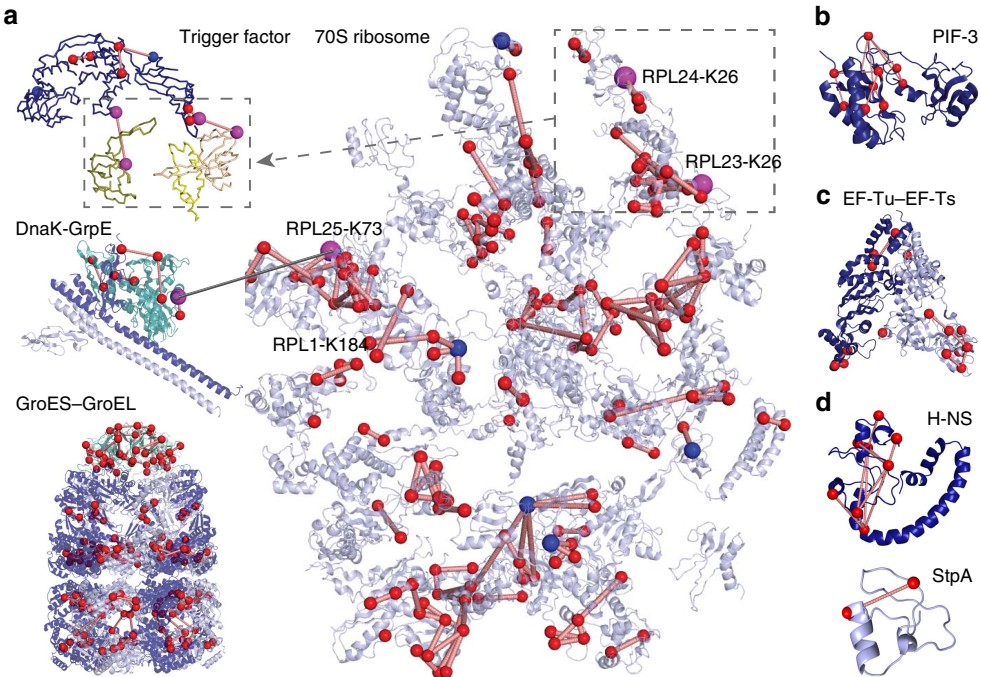

**Figure 3 | Cross-link mapping on representative protein complexes of _E. coli_.** Cross-links shown as red lines could be mapped within one high-resolution structure. These cross-link are within the expected distance range of the DSSO cross-linker (general Cα–Cα distance range: 7–25 Å, see Supplementary Discussion for additional information on the ribosomal cross-links). Lys residues that are cross-linked within a single high-resolution structure are shown as red spheres; Lys residues that are involved in cross-links between two high-resolution structures are shown as purple spheres; Lys residues that exhibit cross-links indicating polysome formation are shown as blue spheres. (**a**) Architecture of chaperone–ribosome co-assemblies (top-left: TF (PDB: 2VRH); middle-left: DnaK-GrpE complex (PDB: 1DKG); bottom-left: GroES-GroEL (PDB: 1PCQ); right: 70S ribosome (PDB: 3JCD)). (**b**) Cross-links mapped onto a homology model of prokaryotic translation initiation factor 3. (**c**) Cross-links confirm the crystal structure of the elongation factor Tu and elongation factor Ts complex (PDB entry 4PC1). (**d**) Cross-links within the bacterial DNA remodeler and transcriptional regulator H-NS (homology model) and its paralogue StpA (PDB entry 2LRX).

translational chaperone-assisted protein folding in prokaryotic cells[14]. Our XL-MS analysis also covered several protein assemblies involved in earlier steps of the protein synthesis pathway (see Supplementary Discussion), for example, translation initiation factors IF-2 (Supplementary Fig. 3a) and IF-3 (Fig. 3b), elongation factors EF-Tu, EF-Ts and EF-G (Fig. 3c and Supplementary Fig. 3b), as well as the transcriptional regulators H-NS and StpA (Fig. 3d).

Another example we highlight focuses on proteins involved in bacterial carbohydrate catabolism, especially the pyruvate dehydrogenase and α-ketoglutarate dehydrogenase complexes (Fig. 4). Both protein assemblies belong to the group of α-keto acid dehydrogenase multi-enzyme complexes, containing up to 24 copies of three enzymes (E1, E2 and E3). These enzymes contain several flexible linker regions since their catalytic action is characterized by large domain movements[15]. Although the catalytic functions are biochemically well characterized, the quaternary structures are not yet fully understood. Intriguingly, the identified cross-links cover parts of the yet uncharacterized flexible linker regions and provide insights into the higher-order structure of both multi-enzyme complexes, as evidenced by cross-links between the pyruvate dehydrogenase E1 and E3 crystal structures, and between the α-ketoglutarate dehydrogenase E1 and E2 crystal structures. Remarkably, several of these cross-links could only be identified with the new data acquisition and analysis strategies (examples are shown in Fig. 2c and Supplementary Fig. 4), demonstrating that the XlinkX v2.0 workflow can provide essential additional information in proteome-wide XL-MS studies. Moreover, the identified cross-links agree with published high-resolution structures of several

other metabolic enzymes (Fig. 4c–g), underscoring that our cross-links reflect native protein contacts (see also Supplementary Discussion).

XL-MS analysis of the HeLa cell lysate using XlinkX v2.0 revealed 3,301 unique Lys-Lys cross-links at 1% FDR and 3,689 cross-links at 2% FDR (Supplementary Data 1). The number of cross-links indicates a more than 80% improvement compared to our previously reported XlinkX v1.0 workflow[9], demonstrating the advantages of implementing mass-difference triggered MS3 acquisitions and the intensity-based precursor determination strategy.

## Discussion

XL-MS is a valuable technique to probe protein structures and interactions _in vitro_ and _in vivo_. The ultimate goal of XL-MS is to chart entire interaction networks at proteome level. Such highly complex XL-MS studies require an appropriate bioinformatics framework. With XlinkX, we aim to provide this framework by facilitating highly confident and efficient crosslink identification against full proteome databases. Here we introduced XlinkX v2.0, which provides several advancements to the proteome-wide XL-MS workflow. First, we improved the confidence of cross-link identification by implementing a hybrid CID–MS2–MS3–ETD–MS2 acquisition strategy that validates cross-links at MS2 and MS3 level. Second, we demonstrated that the results obtained by combining XlinkX v2.0 with a CID–MS2–MS3 acquisition approach are comparable with the results obtained by the previously published CID–ETD–MS2 acquisition strategy[9]. This further extends the applicability of the XlinkX v2.0 workflow to

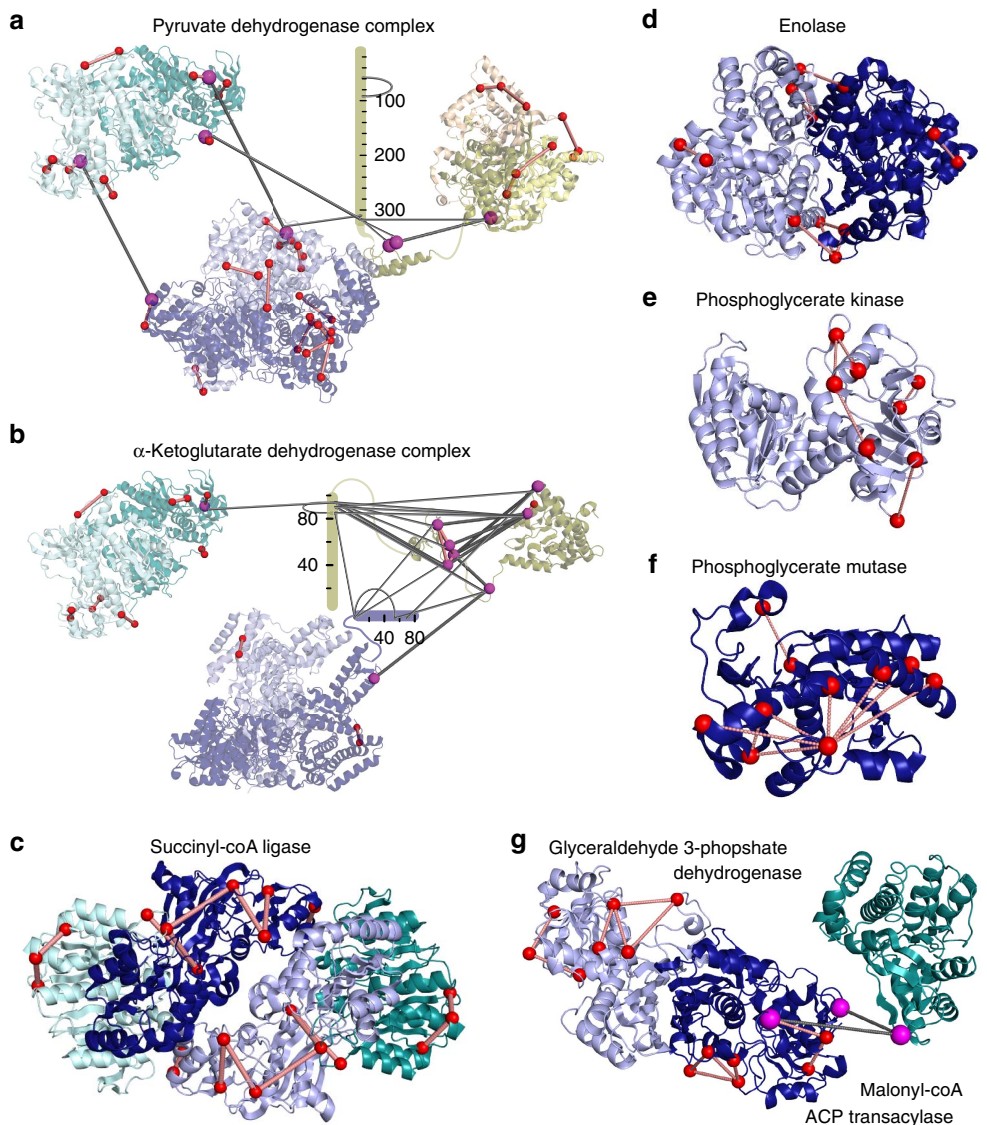

**Figure 4 | Structural insights into enzymes of the carbohydrate catabolism.** Cross-linked Cα atoms are shown as spheres. All cross-links within one crystal structure (shown as red lines) comply with the distance limit of the DSSO cross-linker (Cα- Cα distance range 8–27 Å). Crosslinks representing previously unknown interactions are depicted as gray lines connecting two magenta spheres. (**a,b**) Architecture of α-keto acid dehydrogenase multi-enzyme complexes. Each component is shown in its minimal oligomeric state, as suggested by X-ray crystallography. Structurally uncharacterized regions are shown as sequence bars. (**a**) Pyruvate dehydrogenase. The E1 component (OPD1) is depicted in shades of blue (PDB entry 2G25). The E2 component (ODP2) is shown in shades of yellow as a homo-trimer starting from residue 384 (PDB entry 4N72) and a single copy of the E1-binding E2 region (residues 327–372, PDB entry 4QOY). The E1/E2 interface was determined by structurally aligning the PDB entries 2G25 and 4QOY using Pymol v1.5. The E3 component (DLDH) is shown in shades of teal (PDB entry 4JDR). (**b**) α-Ketoglutarate dehydrogenase. Depicted are an E1 homo-dimer (shades of blue, PDB entry 2JGD), an E2 monomer (yellow, PDB entries 1BAL and 1SCZ) and an E3 homo-dimer (PDB entry 4JDR). To visualize cross-links originating from E2 Lys-156, the C-terminal loop region of PDB entry 1BAL was extended by three residues using the 'rebuild' function of Pymol v1.5. In addition, cross-links were mapped onto structures of (**c**) succinyl-coenzyme A ligase α (light and dark teal)/β(light and dark blue)-hetero-tetramer (PDB entry 1JKJ), (**d**) enolase homo-dimer (PDB entry 1E9I), (**e**) phosphoglycerate kinase (PDB entry 1ZMR), (**f**) phosphoglycerate mutase (PDB entry 1E58) and (**g**) glyceraldehyde 3-phosphate dehydrogenase homo-dimer (light and dark blue, PDB entry 1DC3), which was additionally found to be cross-linked to malonyl-coenzyme A acyl carrier protein transacylase (dark teal, PDB entry 1MLA).

mass spectrometers without ETD fragmentation capabilities. Third, we enhanced the efficiency of cross-link identification by devising an intensity-based precursor mass determination strategy. Thereby, XlinkX v2.0 is now able to uncover cross-links that do not have ideal signature peak patterns in the CID–MS2 spectrum. This feature can be very beneficial when using MS-cleavable crosslinkers that have a higher variation of the signature fragmentation pattern, further expanding the versatility of XlinkX v2.0 and the array of MS-cleavable cross-linkers that are suitable for proteome-wide XL-MS studies.

We benchmarked the XlinkX v2.0 workflow in proteome-wide XL-MS studies using *E. coli* and HeLa whole-cell lysates and identified 1,158 and 3,301 unique Lys-Lys cross-links at ∼1% FDR, respectively (see Supplementary Discussion and Supplementary Fig. 5 for more information on the achieved proteome coverage). Applying the same FDR cut-off to our previously published HeLa data set yields ∼1,800 unique Lys-Lys connections, showing that the XlinkX v2.0 workflow increases the number of identified cross-links by >80%. This improvement is partially attributable to the advanced MS instrumentation used in

the present study, however, we conservatively estimate that the new algorithmic design engenders a 50% higher cross-link identification rate. The identified cross-links provide structurally valid information as we have shown by carefully comparing them to existing high-resolution structures of proteins and protein complexes in *E. coli*. Moreover, the cross-links provide new insights into conformations and interactions of several endogenous *E. coli* protein complexes.

We tested the XlinkX v2.0 workflow using the DSSO cross-linker, which is one of the few commercially available MS-cleavable XL reagents. As mentioned in the introduction, the algorithmic design of XlinkX v2.0 should be compatible with any MS-cleavable cross-linker that exhibits a signature gas phase fragmentation pattern. This is illustrated in Supplementary Fig. 6, showing that XlinkX v2.0 can identify cross-links obtained with the BuUrBu cross-linker[11] in the same way as for DSSO. We envisage that the development and commercialization of more MS-cleavable cross-linkers, ideally with different reactivities and spacer arm lengths, will further extend the scope of proteome-wide XL-MS. The future purview of XL-MS-based *in vivo* protein interaction mapping will also depend on the development of cross-linkers with high cell membrane permeability. DSSO is able to cross cell membranes, however with low efficiency, as indicated by a significant reduction of the number of cross-links. It is noteworthy that the affinity-tagged DSSO-related Azide-A-DSBSO cross-linker has recently been successfully applied to cross-link proteins within intact HEK293 cells[16], showing that *in vivo* XL-MS studies with MS-cleavable cross-linkers may soon be within reach.

## Methods

**Sample preparation and cross-linking.** HeLa or *E. coli* cells were lysed in XL buffer (20 mM Hepes, 150 mM NaCl, 1.5 mM MgCl$_2$, 0.5 mM dithiothreitol pH 7.8) by sonication. Cell lysate (1 mg ml$^{-1}$) was cross-linked with 0.3 mM or 1 mM DSSO cross-linker for 1 h at room temperature and quenched with 20 mM Tris-HCl pH 8.0. Cross-linked proteins were denatured, reduced, alkylated and sequentially proteolysed with Lys-C and trypsin. Protein digests were desalted using Sep-Pak C18 cartridges (Waters), fractionated by SCX and stored at $-20\,^{\circ}$C for further use.

**LC–MS/MS analysis.** The later SCX fractions, which predominantly contain the longer and higher charged peptides ($z > 3$), were analysed using the following: (1) an ultra HPLC Agilent 1,200 system (Agilent Technologies) coupled on-line to an Orbitrap Fusion mass spectrometer (Thermo Fisher Scientific) or (2) an ultra-HPLC Proxeon EASY-nLC 1,000 system (Thermo Fisher Scientific) coupled on-line to an Orbitrap Fusion Lumos mass spectrometer (Thermo Fisher Scientific). Reverse-phase separation was accomplished using a 2 h gradient for *E. coli* fractions or a 3 h gradient for HeLa fractions. Samples were analysed using either of the four acquisition strategies, that is, (1) CID–MS2, (2) CID–ETD–MS2 (sequential CID–MS2 and ETD–MS2 acquisitions on the same MS1 precursor ions), (3) CID–MS2–MS3 (MS3 scans targeting each of the signature peaks in CID–MS2) and (4) CID–MS2–MS3–ETD–MS2 (a combination of 2 and 3). Specifically, in the CID–MS2–MS3–ETD–MS2 approach, sequential CID–ETD–MS2 acquisitions were performed to each MS1 precursor. Subsequently, mass-difference-dependent CID–MS3 acquisitions were triggered if a unique mass difference ($\Delta = 31.9721$ Da) was observed in the CID–MS2 spectrum. MS1 and MS2 scans were acquired in the Orbitrap with a respective mass resolution of 60,000 and 30,000, whereas MS3 scans were acquired in the ion trap. Precursor isolation windows were set to 1.6 *m/z* at MS1 level and 3 *m/z* at MS2 level. The normalized collision energy was set to 25% for CID–MS2 scans and 35% for CID–MS3 scans. Calibrated charge dependent ETD parameters were enabled.

**Data analysis.** The raw data files were converted to *.mgf files using Proteome Discoverer 1.4 software (Thermo Fisher Scientific). The MS2 spectra (CID–MS2 and ETD–MS2) were deconvoluted with the add-on node MS2-Spectrum Processor using default settings. The in-house developed algorithm XlinkX v2.0 was used for the main search. The following settings were used: MS1 precursor ion mass tolerance: 10 p.p.m.; MS2 fragment ion mass tolerance: 20 p.p.m.; MS3 fragment ion mass tolerance, 0.6 Da; fixed modification: Cys carbamidomethylation; variable modification: Met oxidation; allowed number of missed cleavages: 3. All MS2 and MS3 spectra were searched against concatenated target-decoy databases of *E. coli* or *Homo sapiens*. Cross-links were reported at a 1 and 2% FDR, as indicated in the main text and the Supplementary Data 1. For the MS3-only data

analysis, we reconstituted a data analysis pipeline based on the description of the initial XL-MS study using the DSSO cross-linker[10]. In this workflow, the linkage between MS2 precursor ions and MS3 spectra was generated by an in-house developed script using the *.mgf files extracted from Proteome Discoverer 1.4. Subsequently, the MS3 spectra were searched with Sequest using the same parameters as in the XlinkX v2.0 workflow described above. Using an in-house script, the Sequest results were combined to cross-link identifications based on the signature fragmentation pattern of DSSO and the established mass relationship of the cross-linker-cleaved MS2 precursor ions (see equations below).

**Software description.** Compared with the previous version of XlinkX[9], the new version of the software contains two key novel features. First, in the precursor mass determination step, we complemented the $\Delta m$-based precursor mass determination strategy with an intensity-based strategy, where the software determines the precursor mass of each linked peptide based on the masses of the intact cross-link (the MS1 precursor) and only one of the signature peaks (in the CID–MS2 spectrum). In the previously described $\Delta m$-based precursor mass determination strategy, the presence of all four signature peaks was required to distinguish them from the other fragment ions in the same spectrum and to calculate the masses of the two linked peptides. The formulas are described as follows[9]:

$$\Delta m = m_{\mathrm{L}} - m_{\mathrm{S}} = m_{\alpha-\mathrm{L}} - m_{\alpha-\mathrm{S}} = m_{\beta-\mathrm{L}} - m_{\beta-\mathrm{S}}$$
$$m_{\mathrm{prec}} = m_{\alpha-\mathrm{L}} + m_{\beta-\mathrm{S}} = m_{\alpha-\mathrm{S}} + m_{\beta-\mathrm{L}}$$
$$m_{\alpha} = m_{\alpha-\mathrm{L}} - m_{\mathrm{L}} = m_{\alpha-\mathrm{S}} - m_{\mathrm{S}}$$
$$m_{\beta} = m_{\beta-\mathrm{L}} - m_{\mathrm{L}} = m_{\beta-\mathrm{S}} - m_{\mathrm{S}}$$

(where $m_{\alpha}$ and $m_{\beta}$ are the masses of the two linked peptides $\alpha$ and $\beta$, $m_{\mathrm{S}}$ and $m_{\mathrm{L}}$ are the masses of the shorter and longer arm of the cross-linker and $m_{\mathrm{prec}}$ is the mass of the MS1 precursor).

Alternatively, in the intensity-based strategy, we directly take the top $n$ ($n$ is a user-defined parameter) most intense peaks in the CID–MS2 spectrum and postulate that these ions contain at least one of the signature peaks. This assumption is based on the observation that cross-linker cleavage is very likely to generate high abundant product ions. Subsequently, the masses of the two linked peptides are calculated as:

$$m_{\alpha} = m_{\mathrm{h}} - m_{\mathrm{L}} \ \mathrm{OR} \ m_{\alpha} = m_{\mathrm{h}} - m_{\mathrm{S}}$$
$$m_{\beta} = m_{\mathrm{prec}} - m_{\alpha} - m_{\mathrm{L}} - m_{\mathrm{S}}$$

(where $m_{\mathrm{h}}$ is the mass of the selected MS2 fragment ion).

Of note, the precursor masses of both linked peptides calculated in this way will be incorrect if the selected MS2 fragment ion is not a signature peak. However, by using a stringent fragment ion score cutoff for both linked peptides, these results can be confidently excluded from the true cross-link identifications.

The second novel feature of XlinkX v2.0 is to analyse data from the combined MS2–MS3 fragmentation strategy. In the product ion matching step, the software compares the theoretical fragment ions of the candidate peptides first with MS2 data (including CID–MS2 and ETD–MS2) and subsequently with the MS3 spectra for each of the linked peptides. XlinkX uses a probability scoring algorithm (see also refs 9,17) to calculate the confidence of each candidate sequence for MS2 and MS3 spectra, respectively, which is reported as $n$-score in the result. It is calculated by

$$n\text{-score} = \left(1 - \sum_{i=0}^{n-1} \frac{e^{-xf}(xf^{i})}{i!}\right) * N$$

(where $n$ is the number of matching fragments, $x$ is the probability of an observed fragment ion matching a random theoretical fragment ion by chance, $f$ is the total number of theoretical fragment ions for each linked peptide and $N$ is the number of proteins in the database).

The MS3 score needs to pass the user-defined MS3 score cutoff to be considered for final score calculation. The final score of each linked peptide is a combination of MS2 and MS3 scores, which is measured by:

$$n\text{-score}(\alpha) = n\text{-score}(\mathrm{MS2}\alpha) \times n\text{-score}(\mathrm{MS3}\alpha)$$
$$n\text{-score}(\beta) = n\text{-score}(\mathrm{MS2}\beta) \times n\text{-score}(\mathrm{MS3}\beta)$$

(where $n$-score($\alpha$) is the $n$-score for peptide $\alpha$, whereas $n$-score($\beta$) is the $n$-score for peptide $\beta$).

Finally, XlinkX employs a target-decoy strategy for FDR estimation using a concatenated target and decoy database.

**Structural representation of cross-links.** Cross-links were mapped onto published high-resolution structures of *E. coli* proteins (PDB codes indicated in the text). If *E. coli* protein structures were not available, homology models were created using the Phyre2 web server[18]. The use of homology models is indicated in the text. It was ascertained that the model confidence is >90% for more than 90% of the residues.

**Data availability.** All cross-links are reported in the Supplementary Data. XlinkX v2.0 is publically available via http://xlinkx2beta.hecklab.com/. The data that

support the findings of this study are available from the corresponding author upon request.

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

## Acknowledgements

We are indebted to Gert E. Folkerts (Utrecht University) for providing the *E. coli* cell culture facilities. We also thank Chris Etienne (Thermo Fisher Scientific, Rockford IL) for synthesizing the BuUrBu and DSSO cross-linker. This work was partly supported by Proteins@Work, a programme of the Netherlands Proteomics Centre financed by Netherlands Organisation for Scientific Research (NWO) as part of the National Roadmap Large-scale Research Facilities of the Netherlands (project number 184.032.201). This project received also funding from the European Union's Horizon 2020 research and innovation programme (grant agreement MSMed number 686547) and 7th framework programme (grant agreement Manifold number 317371).

## Author contributions

F.L. and A.J.R.H. designed the research. F.L., P.L. and R.V. performed the experiments. F.L. developed XlinkX v2.0. F.L. and P.L. analysed the data. F.L., P.L. and A.J.R.H. wrote the paper. All authors contributed by critically reading the paper.

## Additional information

**Competing interests:** R.V. is an employee of Thermo Fisher Scientific, developer and distributor of the Orbitrap Fusion/Lumos. The remaining authors declare no competing financial interests.

**Publisher's note**: 

