## [Peer Review File · Nature Communications]

Editorial Note: This manuscript has been previously reviewed at another journal that is not operating a transparent peer review scheme. This document only contains reviewer comments and rebuttal letters for versions considered at Nature Communications. Mentions of prior referee reports have been redacted.

PEER REVIEW FILE

Reviewers' Comments:

Reviewer #2 (Remarks to the Author):

Reviewer's report on “Optimized fragmentation schemes and data analysis strategy for proteome-wide cross-link identification” by Liu et al., submitted to Nature Communications.

The authors have submitted a revised version of the manuscript, which I reviewed in its original form for Nature Methods, to Nature Communications together with a point-by-point reply addressing the issues that I raised in my report for Nat. Meth.

The authors have included another dataset derived from cross-linking of entire HeLa cell lysates in the current manuscript. I have this point in my previous report.

Regarding the point-by-point-reply, the authors have addressed my specific concerns, except for the 2% FDR applied, which I still regard as inadequate (and a search with 1% FDR and showing the respective results could have clarified this issue – or not).

All in all, I would like to state that I am in favour of the approach of gas-phase-cleavable protein–protein cross-linkers and their Implementation within a reliable software tool and the application on a very complex sample (entire cell lysates). It will be also very interesting to evaluate, once available for the community, whether cleavable and non-cleavable cross-linkers perform differently or equally well.

However, I am still uncertain as to whether this work in its present form is suitable for publication in Nat. Comm. Irrespective of the that I raised concerning their first version (submitted to Nat. Meth.), which I considered unsuitable for publication in Nat Meth. and hence did to go into more detailed constructive criticism, there are some issues that should be addressed before I am convinced that this current manuscript is beyond being a preliminary proof-of-

principle study.

In this connection: (i) The authors state that their software can be applied to other cleavable cross-linkers. Here, data are missing. Should it not have been shown by a proof-of-principle study? Or will this then simply result in another manuscript? (ii) The authors discuss the significance of their data in a biological context in the supplementary data. However, should it not be the sense of cross-linking on complex systems that intact cells (or their organelles) can be cross-linked? Cross-linking of lysate is always less reliable. Of course, the application of the workflow described here is independent of this; however, the corresponding interpretation of the data in a biological context might not be. In other words, is the DSSO membrane permeable, and can intact cells be cross-linked? Application of the cleavable cross-linker to intact cells would have been beneficial. Otherwise, cleavable cross-linkers will remain only suitable in situations where the membrane is permeabilised, or for isolated macromolecular assemblies.

As I indicate above, I am not completely against this work, but had hoped – in particular for a high-impact journal – for more improvements as compared with their original version.

Reviewer #3 (Remarks to the Author):

The manuscript „Optimized fragmentation schemes and data analysis strategies for proteome-wide cross-link identification“ from the Heck group reports the XlinkX v2 strategy for the identification of lysine-lysine crosslinks in complex samples using the cleavable crosslinker DSSO.

This study highlights in particular two improvements of a previously published workflow increasing the number of crosslinks identified from lysates by 50%. The use of a Orbitrap Fusion Lumos Tribrid instrument facilitated the targeted acquisition of MS3 spectra from peptide precursors of the signature peptides derived by crosslinker cleavage. In addition a hybrid method including simultaneous ETD MS2 spectra resulted in even higher numbers of identified crosslinks. The XlinkX v2 software is designed for the analysis of this type of mass spectrometric crosslink experiments. The current software version also implements an intensity-based precursor mass determination strategy accounting for low-abundant or missing precursor peptides and thereby identifying previously undetectable crosslinks. This study nicely demonstrates the technical improvements of a reported workflow which will most likely become integral parts of future crosslinking studies in complex samples.

However, the reported increase in sensitivity is still moderate, the highlighted structural biological insights are demonstrated on highly abundant E.coli proteins and thus the novel

biological insights gained by the improved workflow do not go beyond the insights of the original paper in Nature Methods (2015). Also the claim that CID MS2 MS3 and hybrid fragmentation strategies may allow use of different cleavable crosslinkers has not been demonstrated. Although it is a very nice methodological paper I am a bit reluctant to agree to its publication in Nature communications.

Overall, I wonder whether technical improvements of a workflow that have not yet resulted in significantly "deeper" insights justify publication of this manuscript in Nature Communications and if a journal more specialized on method development would be a better choice.

Response to reviewer 2

Comment: Reviewer's report on "Optimized fragmentation schemes and data analysis strategy for proteome-wide cross-link identification" by Liu et al., submitted to Nature Communications. The authors have submitted a revised version of the manuscript, which I reviewed in its original form for Nature Methods, to Nature Communications together with a point-by-point reply addressing the issues that I raised in my report for Nat. Meth. The authors have included another dataset derived from cross-linking of entire HeLa cell lysates in the current manuscript. I have this point in my previous report.

Response: Thank you for your positive feedback.

Comment: Regarding the point-by-point-reply, the authors have addressed my specific concerns, except for the 2% FDR applied, which I still regard as inadequate (and a search with 1% FDR and showing the respective results could have clarified this issue – or not).

Response: We have reanalyzed our data and now report the results of both 1% and 2% FDR. In summary, the number of crosslinks are 1,158 unique Lys-Lys cross-links at 1% FDR and 1,330 cross-links at ~2% FDR for *E. coli* and 3,301 cross-links at 1% FDR and 3,689 cross-links at 2% FDR for HeLa.

Comment: All in all, I would like to state that I am in favour of the approach of gas-phase-cleavable protein–protein cross-linkers and their Implementation within a reliable software tool and the application on a very complex sample (entire cell lysates). It will be also very interesting to evaluate, once available for the community, whether cleavable and non-cleavable cross-linkers perform differently or equally well. However, I am still uncertain as to whether this work in its present form is suitable for publication in Nat. Comm. Irrespective of the that I raised concerning their first version (submitted to Nat. Meth.), which I considered unsuitable for publication in Nat Meth. and hence did to go into more detailed constructive criticism, there are some issues that should be addressed before I am convinced that this current manuscript is beyond being a preliminary proof-of-principle study. In this connection: (i) The authors state that their software can be applied to other cleavable cross-linkers. Here, data are missing. Should it not have been shown by a proof-of-principle study? Or will this then simply result in another manuscript? (ii) The authors discuss the significance of their data in a biological context in the supplementary data. However, should it not be the sense of cross-linking on complex systems that intact cells (or their organelles) can be cross-linked? Cross-linking of lysate is always less reliable. Of course, the application of the workflow described here is independent of this; however, the corresponding interpretation of the data in a biological context might not be. In other words, is the DSSO membrane permeable, and can intact cells be cross-linked? Application of the cleavable cross-linker to intact cells would have been beneficial. Otherwise, cleavable cross-linkers will remain only suitable in situations where the membrane is permeabilised, or for isolated macromolecular assemblies.

Response: The initial version of XlinkX has already been applied in combination with other MS-cleavable cross-linkers after it was published in 2015 (see Arlt et al., *Anal Chem*, 2016), showing that the XlinkX principle is not limited to DSSO. However, we agree with the reviewer that this should also be demonstrated for XlinkX v2.0. We chose to illustrate the

versatility of XlinkX v2.0 using the BuUrBu cross-linker. Our example (Page 10 in revised manuscript and Supplementary Figure 9 in revised Supplementary Materials) shows that XlinkX v2.0 is able to interpret BuUrBu data the same way as with DSSO. Regarding the second the point of the reviewer, “*cross-linking of lysate is always less reliable*”, we would like to respectfully disagree with this general statement. *In vitro* biochemistry studies have been always of significant importance in characterizing chemical and physical properties of proteins and many indispensable techniques have been developed based on cell lysate setups. In fact, *in vitro* cross-linking (i.e., cross-linking of protein samples derived from cell lysates) has made significant contributions over the past years in the fields of structural biology and cell biology. Illustrative examples include structural investigations of many macromolecular assemblies, such as the mammalian mitochondrial ribosome (Greber et al., *Nature*, 2014), the nuclear pore complex (Shi et al., *Mol Cell Proteomics*, 2014), the Pol II-mediator initiation complex (Plaschka et al., *Nature*, 2015 and Robinson et al., *Cell*, 2016). Furthermore, cross-linking has also been applied in combination with affinity purification (which is also on the basis of cell lysates) and obtained significant biological insights of PP2A interaction network (Herzog et al., *Science*, 2012). However, we agree with the reviewer that using membrane permeable cross-linkers can be beneficial as it may preserve transiently formed or less stable *in vivo* protein interactions. In this regard, a recent publication reported the development of a DSSO-related Azide-A-DSBSO cross-linker (MS cleavable) and successfully showed its ability to penetrate the membrane of HEK293 cells (Kaake et al., *Mol Cell Proteomics*, 2014). This demonstrates that *in vivo* cross-linking with MS-cleavable (and therefore XlinkX-compatible) cross-linkers is feasible. We have added this information to the Discussion section of our revised manuscript.

As I indicate above, I am not completely against this work, but had hoped – in particular for a high-impact journal – for more improvements as compared with their original version.

Response to reviewer 3

Comment: The manuscript „Optimized fragmentation schemes and data analysis strategies for proteome-wide cross-link identification“ from the Heck group reports the XlinkX v2 strategy for the identification of lysine-lysine crosslinks in complex samples using the cleavable crosslinker DSSO. This study highlights in particular two improvements of a previously published workflow increasing the number of crosslinks identified from lysates by 50%. The use of a Orbitrap Fusion Lumos Tribrid instrument facilitated the targeted acquisition of MS3 spectra from peptide precursors of the signature peptides derived by crosslinker cleavage. In addition a hybrid method including simultaneous ETD MS2 spectra resulted in even higher numbers of identified crosslinks. The XlinkX v2 software is designed for the analysis of this type of mass spectrometric crosslink experiments. The current software version also implements an intensity-based precursor mass determination strategy accounting for low-abundant or missing precursor peptides and thereby identifying previously undetectable crosslinks. This study nicely demonstrates the technical improvements of a reported workflow which will most likely become integral parts of future crosslinking studies in complex samples.

Response: Thank you for the positive feedback.

Comment: However, the reported increase in sensitivity is still moderate, the highlighted structural biological insights are demonstrated on highly abundant E.coli proteins and thus

the novel biological insights gained by the improved workflow do not go beyond the insights of the original paper in Nature Methods (2015).

Response: As also indicated by the reviewer the focus of our manuscript is on the advancement of MS acquisition and data analysis strategies to enable more confident cross-link identification from highly complex samples. We believe the best way to demonstrate a methodological advancement is to compare the performance to existing approaches using the same type of sample. Based on the number of cross-links identified from a HeLa cell lysate, we have shown that the new XlinkX v2.0 workflow identifies 80% more cross-links than the initial XlinkX workflow. We find this a major gain. For clarity, we discuss this improvement more explicitly in the revised manuscript (page 9-10).

Evidently, the improved cross-link identification rate brings about a deeper proteome coverage. Regarding the reviewer's concern that we only cover highly abundant *E.coli* proteins, we would like to mention that the proteome coverage of our cross-links is thoroughly discussed in the Supplementary Material, where we state that, "we show 10%, 35% and 64% of proteins involved in cross-links are among the top 5%, 10% and 30% most abundant proteins, respectively, based on the rankings from iBAQ values obtained by parallel shot-gun proteomics experiments. Furthermore, 39%, 60%, 74%, 83% and 88% of cross-links are detected within the top 10%, 20%, 30%, 40% and 50% most abundant proteins, respectively." Although these results indicate cross-links are enriched in proteins with higher abundances, we do cover approximately the 400 most abundant *E.coli* proteins in our cross-linking dataset, which presents one of the most comprehensive crosslinking dataset of *E. coli* so far. Illustrative examples of new biological insights are also provided in Figure 3, showing for example cross-links in structurally uncharacterized regions of pyruvate dehydrogenase and α -ketoglutarate dehydrogenase complexes, but it is beyond the scope of this work to discuss all these individual findings in this manuscript.

Comment: Also the claim that CID MS2 MS3 and hybrid fragmentation strategies may allow use of different cleavable crosslinkers has not been demonstrated. Although it is a very nice methodological paper I am a bit reluctant to agree to its publication in Nature communications.

Response: As mentioned in our response to reviewer 2, we have added a proof-of-principle experiment using another cleavable cross-linker (i.e., BuUrBu) in our revised manuscript. We hope that take away this concern.

Overall, I wonder whether technical improvements of a workflow that have not yet resulted in significantly "deeper" insights justify publication of this manuscript in Nature Communications and if a journal more specialized on method development would be a better choice.

With cross-linking become rapidly very popular and the previous version of XlinkX described in our Nature Methods paper already being quite a work of impact, we believe this version and new fragmentation schemes derived will be quickly adopted by many in the field. Therefore, we think Nature Communications is the right venue for this work, but evidently that is an editorial decision.

Reviewers' Comments:

Reviewer #2 (Remarks to the Author):

Reviewer's report on the revised manuscript "Optimized fragmentation schemes and data analysis strategies for proteome-wide cross-link identification" by Liu et al., submitted to Nature Communications.

In their revised version and in their rebuttal letter the authors have addressed most of the points that I raised in my review of the original manuscript. These were principally the 'false discovery' rate used and the application of a different gas-phase-cleavable cross-linker, i.e. BuUrBu.

I also raised some concern as to whether the cross-linking of cellular lysate is appropriate, or whether cross-linking should not rather be applied to intact cells, in particular when novel protein-protein interactions are to be identified. In their rebuttal letter, the authors list several examples of the validity of protein-protein cross-links of isolated macromolecular assemblies as well as in a large interactome study of affinity-purified complexes. However, these cross-links were derived from isolated complexes, and hence the data were interpreted with respect to the isolated complexes. Nevertheless, in the present proteome-wide study by Liu et al. the authors found several novel protein-protein interactions after cross-linking. It is still my belief that one has to be very careful in interpreting such data, as until now no corresponding complexes have been isolated from cells, and one has to bear in mind that lysis of cells might induce artificial protein-protein cross-links. In this respect, evidence for the protein-protein interaction identified *ex vivo* should also be found *in vivo*. However, I am aware that the cross-linker used here might not yet meet the requirements for application *in vivo* (, but have the authors ever tested for its *in vivo* application?). In this context, I welcome the fact that the authors did discuss this possibility.

All in all – and in particular because the application of this kind of cell-wide cross-linking is of interest for the broad community of molecular-biological and cell-biological researchers – I consider that the revised version of the manuscript should be accepted for publication in Nat. Comm.

Reviewer #3 (Remarks to the Author):

The authors have sufficiently addressed my comments and I agree with publication of this manuscript in Nature Communications.